# Origin of Ciguateric Fish: Quantitative Modelling of the Flow of Ciguatoxin through a Marine Food Chain

**DOI:** 10.3390/toxins14080534

**Published:** 2022-08-03

**Authors:** Michael J. Holmes, Richard J. Lewis

**Affiliations:** Institute for Molecular Bioscience, The University of Queensland, Brisbane 4072, Australia; m.holmes@imb.uq.edu.au

**Keywords:** ciguatera, ciguatoxin, *Gambierdiscus*, Spanish mackerel, *Scomberomorus commerson*, Platypus Bay

## Abstract

To begin to understand the impact of food chain dynamics on ciguatera risk, we used published data to model the transfer of ciguatoxins across four trophic levels of a marine food chain in Platypus Bay, Australia. The data to support this first attempt to conceptualize the scale of each trophic transfer step was limited, resulting in broad estimates. The hypothetical scenario we explored generated a low-toxicity 10 kg Spanish mackerel (*Scomberomorus commerson*) with a flesh concentration of 0.1 µg/kg of Pacific-ciguatoxin-1 (P-CTX-1, also known as CTX1B) from 19.5–78.1 µg of P-CTX-1 equivalents (eq.) that enter the marine food chain from a population of 12–49 million benthic dinoflagellates (*Gambierdiscus* sp.) producing 1.6 × 10^−12^ g/cell of the P-CTX-1 precursor, P-CTX-4B. This number of *Gambierdiscus* could be epiphytic on 22–88 kg of the benthic macroalgae (*Cladophora*) that carpets the bottom of much of Platypus Bay, with the toxin transferred to an estimated 40,000–160,000 alpheid shrimps in the second trophic level. This large number of shrimps appears unrealistic, but toxic shrimps would likely be consumed by a school of small, blotched javelin fish (*Pomadasys maculatus*) at the third trophic level, reducing the number of shrimps consumed by each fish. The Spanish mackerel would accumulate a flesh concentration of 0.1 µg/kg P-CTX-1 eq. by preying upon the school of blotched javelin and consuming 3.6–14.4 µg of P-CTX-1 eq. However, published data indicate this burden of toxin could be accumulated by a 10 kg Spanish mackerel from as few as one to three blotched javelin fish, suggesting that much greater amounts of toxin than modelled here must at certain times be produced and transferred through Platypus Bay food chains. This modelling highlights the need for better quantitative estimates of ciguatoxin production, biotransformation, and depuration through marine food chains to improve our understanding and management of ciguatera risk.

## 1. Introduction

Spanish mackerel (*Scomberomorus commerson*) are pelagic fish targeted by both commercial and recreational fishers [1], despite being one of the highest risk species for ciguatera on the east coast of Australia [2,3,4,5]. Spanish mackerel from the east coast of Australia were previously considered sustainably fished but maximally exploited [6]; however, a recent analysis has suggested that the stock has fallen to ~17% of the biomass that existed prior to commercial fishing [7]. The source of ciguatoxins (CTX) contaminating many of the toxic Spanish mackerel caught in southern Queensland waters includes Platypus Bay, a small bay on the lee side of K’gari (Fraser Island) [8]. We recently developed a conceptual model for the bioaccumulation of CTX through the Platypus Bay food chain into Spanish mackerel and its subsequent dilution through growth and depuration [5]. CTX originate from *Gambierdiscus* growing epiphytically on free-living macroalgae (*Cladophora* sp.) carpeting the bottom of much of Platypus Bay and are thought to be transferred through three trophic levels into Spanish mackerel [5,8,9,10]:

*Gambierdiscus* → invertebrates → blotched javelin fish (*Pomadasys maculatus*) → Spanish mackerel.

The major CTX analog accumulated by Spanish mackerel to cause ciguatera on the east coast of Australia is Pacific-ciguatoxin-1 (P-CTX-1) [11,12], also known as CTX1B [13]. In this paper we model the transfer of CTX through the Platypus Bay food chain and estimate the number of CTX-producing *Gambierdiscus* required to produce a Spanish mackerel with a muscle (flesh) concentration of 0.1 µg/kg P-CTX-1. This flesh concentration would likely produce mild symptoms in 2 out of 10 people [14] and is 10-fold higher than the USFDA recommended safe limit for consumption of seafood [15]. We model the accumulation of P-CTX-1 into 10 kg Spanish mackerel as this is the maximum size accepted for sale by the Sydney Fish Market because of the perceived risk of ciguatera from larger specimens [16]. Anecdotal information suggests that many recreational anglers also prefer fish <10 kg, because of perceived better eating qualities of smaller fish and/or their perception that larger fish are a greater risk of causing ciguatera. Spanish mackerel can grow to ~70 kg [17] although fish of this size are rarely caught now, with the average size caught from the east coast of Queensland now being <8 kg (see Buckworth et al. [1], O’Neill et al. [6], and Tanimoto et al. [7] for details about the fishery).

In this paper, we quantitatively model the transfer of CTX between four trophic levels of Platypus Bay, an ecosystem for which we previously developed a conceptual model [5]. We developed our model by choosing a target CTX concentration for the flesh of Spanish mackerel and then back calculating the quantity of toxin required to be transferred through each trophic level to cause this level of contamination. Our approach can help future researchers identify key knowledge gaps to improve understanding of how CTX flows through marine food chains to cause human poisoning. This model suggests that large cell populations of CTX-producing *Gambierdiscus* are required to produce a single toxic fish, which along with the capacity for fish to lose toxicity over time through depuration [5], explains the relative rarity of ciguatera in Australia.

## 2. Results and Discussion

The total burden of CTX accumulated by a Spanish mackerel was estimated for a target flesh concentration of 0.1 µg/kg P-CTX-1, with “CTX” here used to cover all toxic CTX precursors and metabolites. The amount of CTX contaminating fish tissues arises from a dynamic process that changes over time and is dependent on bioaccumulation efficiency and depuration rates, as well as bio-transformations that change toxicity and tissue distribution [18,19,20,21,22,23,24,25]. For example, fish continue to accumulate toxin in muscle after they cease feeding on the toxin source as shown for Pacific surgeonfish [22] and Caribbean pinfish [25], with pinfish muscle reaching ~10% of the total CTX burden after 20 days feeding on pellets spiked with Caribbean-CTX (C-CTX), and ~40% more than 2 months after C-CTX feeding stopped [25]. Using these percentages, we estimated a range for total CTX burden (Table 1), assuming toxin accumulation occurs over a short feeding period (days–weeks), before CTX depuration could significantly reduce the toxin concentration in flesh (see Holmes et al. [5]).

The relationship (r^2^ = 0.98) between whole fish and fillet weight (kg) for Spanish mackerel was derived by Mackie et al. [26]:(1)$$\mathrm{Whole}\;\mathrm{weight}=\left(\mathrm{fillet}\;\mathrm{weight}\times 1.62\right)-0.06$$

We used this equation to estimate the flesh (muscle) weight of a 10 kg Spanish mackerel and back calculate a range for the toxin burden for the fish based upon the targeted flesh concentration of 0.1 µg/kg of P-CTX-1 eq. (Table 1). This estimation for the total toxin burden is simplistic given the dynamic processes occurring [19,20,21,22,23,24,25,27], which may be different between species. However, it provides a starting point that can be refined in the future as additional data becomes available. It is also possible that the relative fraction of toxin contaminating muscle could exceed 40% over longer time frames than those explored by Bennett and Robertson [25]. We assume the remaining carcass tissues (excluding liver and remaining viscera, see Bennett and Robertson [25]) not accounted for by these calculations (mostly bones) contribute minimally to the toxin burden of the fish.

Based upon our model, a 10 kg Spanish mackerel contaminated with a flesh concentration of 0.1 µg/kg P-CTX-1 would have a flesh burden of 0.62 µg of P-CTX-1, and a total fish burden of between 1.6–6.2 µg of P-CTX-1 equivalents (eq.) (Table 1). The latter are the minimum amounts of P-CTX-1 eq. that could be derived from the base of the food chain, as they do not account for losses through each trophic level from the *Gambierdiscus* producing the toxin.

Bennett and Robertson [25] estimated an assimilation rate of 43% for pinfish eating fish pellets contaminated with C-CTX. This transfer efficiency is comparable to that of mullet eating fish pellets spiked with *G. polynesiensis* (42% [20]) but much higher than estimated for juvenile grouper (<7% [23]), and freshwater goldfish (<5% [28]). We used three possible assimilation rates (6%, 43%, and 100%) to estimate the quantity of P-CTX required at each trophic level of Platypus Bay to contaminate Spanish mackerel flesh with 0.1 µg/kg P-CTX-1 (Table 2).

Assuming CTX transfers through three trophic levels before contaminating Spanish mackerel in Platypus Bay, and the best available estimate of assimilation efficiency of 43% for each trophic transfer, suggests that 19.5–78.1 µg of P-CTX-1 eq. is consumed from *Gambierdiscus* at the base of the food chain to produce a lowly toxic, 10 kg Spanish mackerel with flesh contaminated with 0.1 µg/kg P-CTX-1 (Table 2, Figure 1). In comparison, a 6% transfer efficiency [23] requires unrealistically high toxin levels to be passed through trophic levels 1 and 2 (Table 2). A 100% transfer efficiency only requires 1.6–6.2 µg of CTX to be transferred across trophic levels; however, a scenario without toxin loss is not possible. Of note is the much lower transfer efficiency reported for the less toxic analogs P-CTX-2 and -3 (52-epi-54-deoxyCTX1B and 54-deoxyCTX1B [13], respectively) in juvenile grouper [23] than the 6% we have simulated here for P-CTX-1. It is likely that the true transfer efficiency varies between each trophic level for different toxin analogs, but we have no data to incorporate such variation in our model. Although speculative, to the best of our knowledge, this is the first attempt to estimate the levels of toxin production required to produce a poisonous fish in the wild, which may be a useful approach for designing risk management strategies in ciguateric regions. Initial inspection of the modelled amounts of P-CTX-1 eq. produced in the first trophic level to cause a poisonous fish appears to be considerable (19.5–78.1 µg P-CTX, Table 2); however, this range is small compared to what some reefs in the Republic of Kiribati must be able to produce, where moray eels (high trophic level carnivores) can be contaminated with >80 µg/kg of P-CTX-1 eq. [18,19]. It is possible that food chains with fewer trophic links are more efficient for bioaccumulation of CTX into carnivorous fishes. Certainly, the feeding habits of the animals in the ciguateric food chain will affect the type and amount of prey tissue consumed across each trophic level and therefore the rate for toxin transfer. The relatively large size of predators relative to prey in the two apex trophic levels for Platypus Bay (Spanish mackerel feeding on blotched javelin fish, and blotched javelin feeding upon shrimps), suggest that prey will often be eaten whole with little tissue loss. However, crustaceans often process food externally, which may offer greater scope for loss of toxin from the second trophic level.

We next use our estimates for the P-CTX-1 eq. transferred across Platypus Bay trophic levels (Table 2) as targets to explore possible population sizes and scenarios for the *Gambierdiscus*, shrimps, and blotched javelin fish in the food chain transfer of CTX to produce a mildly toxic 10 kg Spanish mackerel (Table 3).

### 2.1. Trophic Level 1 (Gambierdiscus)

*Gambierdiscus* growing on *Cladophora* in Platypus Bay are the presumed origin of ciguatoxins contaminating some of the ciguateric Spanish mackerel caught from the east coast of Australia [5]. However, the *Gambierdiscus* species from Platypus Bay have not yet been identified as previous studies occurred when the genus was known only from a single species (*G. toxicus*). Platypus Bay *Gambierdiscus* were estimated to produce an average of 1.3 × 10^−5^ mouse units (MU) of CTX/cell from the most toxic sample collected from a limited sampling program between 1988–1990 [8]. The CTX analogs produced by some Australian species of *Gambierdiscus* are presumed to include P-CTX-4B (CTX4B [13]) as this, and its stereoisomer P-CTX-4A (CTX4A [13]), are the precursors of P-CTX-1 [9,13,21,29], the major CTX contaminating ciguateric fishes in Australia, including Spanish mackerel [9,11,12,30,31]. However, the identities and relative amounts of the ciguatoxins produced by Australian species of *Gambierdiscus* are still unknown [5]. For this scenario, we modelled the toxicity of Platypus Bay *Gambierdiscus* as being due to P-CTX-4B, which has ~1/20th the potency of P-CTX-1 with a mouse LD_50_ of 5.9 µg/kg, i.e., 1 MU (based upon a 20 g mouse) ≈ 0.12 µg [13,29]. This would equate to Platypus Bay *Gambierdiscus* producing ~1.6 × 10^−12^ g P-CTX-4B/cell. This is a high CTX concentration, but within the range of concentrations reported for CTX analogs from *Gambierdiscus polynesiensis* [32,33,34]. Extraction of toxins from cultured Platypus Bay *Gambierdiscus* were lower than from wild cells at 8.8 × 10^−7^ MU/cell for the major of two CTX analogs detected by mouse bioassay [35]. If this major CTX-analog was P-CTX-4B, it would equate to 1.1 × 10^−13^ g P-CTX-4B/cell.

Based upon our scenario for production of 1.6 × 10^−12^ g P-CTX-4B/cell, and a 1:1 conversion of this analog to P-CTX-1, it would take the food chain accumulation of CTX from 12–49 million Platypus Bay *Gambierdiscus* to produce a 10 kg lowly toxic Spanish mackerel (Table 3). *Gambierdiscus polynesiensis* produces more CTX than any other known *Gambierdiscus* or *Fukuyoa* species [13]. However, this is mostly due to production of P-CTX3C (CTX3C [13]) analogs [33,34], which have not yet been found from Australian *Gambierdiscus* or fish, and have a structurally different backbone (‘E’ ring) to the P-CTX-1 family of toxins, precluding their conversion to P-CTX-1 in the marine food chain [21]. In contrast, P-CTX-4A is a stereoisomer of P-CTX-4B also produced by *G. polynesiensis* [34,36,37], that is ~3.5-fold more toxic to mice than P-CTX-4B [13], which can also bio-transform in the food chain to P-CTX-1 [9,13,21]. If the toxicity found by Holmes and Lewis [35] and Holmes et al. [8] for Platypus Bay *Gambierdiscus* was due to P-CTX-4A rather than P-CTX-4B, then based upon relative toxicities [13], ~3.5-fold more *Gambierdiscus* than modelled here (Table 3) would be required to produce a lowly toxic 10 kg Spanish mackerel (i.e., 42–169 million *Gambierdiscus*).

Darius et al. [34] recently found that P-CTX-4B was the major of the two stereoisomers (P-CTX-4A and -4B) produced by *G. polynesiensis*, although previous studies suggest this can vary [13,33]. However, the maximum concentration of P-CTX-4B so far found from French Polynesian *G. polynesiensis* is only ~0.4 × 10^−12^ g/cell, with maximum combined concentrations of P-CTX-4A and-4B of ~0.6 × 10^−12^ g/cell [34]. This combined cellular concentration is ~2.7-fold less than we have used for our model from Platypus Bay *Gambierdiscus* (1.6 × 10^−12^ g P-CTX-1 eq./cell). If we based our Platypus Bay modelling on the maximum combined P-CTX-4A and-4B concentrations found by Darius et al. [34], then ~3-fold more *Gambierdiscus* than modelled here for Platypus Bay (Table 3), would be required to produce a lowly toxic 10 kg Spanish mackerel (i.e., 33–130 million *Gambierdiscus*).

### 2.2. Trophic Level 2 (Invertebrates)

Invertebrates, principally small alpheid shrimps, living within the *Cladophora* are a presumed intermediate for the transfer of CTXs from epiphytic *Gambierdiscus* into the carnivorous blotched-javelin fish (*P. maculatus*) in Platypus Bay [5,10]. Lobsters are the only other crustaceans so far suggested to accumulate CTX [19,38]. However, crustaceans such as lobsters and alpheid shrimps may not feed directly on microalgae which could indicate additional links in the Platypus Bay food chain leading to ciguatera in Spanish mackerel. The most obvious alternate food chain for transfer of ciguatoxins in Platypus Bay is through the dominant herbivorous fish (rabbitfish), but these are apparently non-toxic [5,39]. The ciguatoxicity of a bulk sample of shrimps (42.5 g, wet weight) collected from Platypus Bay during May 1991 was 0.5 MU [10]. If this toxicity was mostly due to P-CTX-4B (i.e., before the biotransformation to the more toxic P-CTX-1), the sample would contain a total of ~6.0 × 10^−8^ g of P-CTX-4B. This was extracted from ~280 shrimps with an average size of 0.15 g [10], which for this scenario suggests an average toxin content per shrimp of 2.1 × 10^−10^ g of P-CTX-4B. Assuming an assimilation efficiency of 100% either directly from *Gambierdiscus* or through an additional unknown food chain link would indicate an “average” shrimp would have accumulated this amount of toxin from ~131 *Gambierdiscus* cells that were producing 1.6 × 10^−12^ of P-CTX-4B/cell. The true assimilation efficiency for the trophic transfer of CTX’s into invertebrates is not known but this would increase to ~305 cells if an assimilation efficiency of 43% [25] was assumed with shrimp directly feeding on *Gambierdiscus.* CTX was below the detection limit in later sampling of shrimp populations from Platypus Bay *Cladophora*, indicating toxicity had declined after the May 1991 sampling [10]. However, it is possible that shrimps may sometimes accumulate greater levels of CTX than the toxin levels found by Lewis et al. [10] and used in our model. In contrast, aquarium experiments have shown that giant clams (*Tridacna maxima*) only accumulate a small amount (0.6%) of the toxin load from suspended *Gambierdiscus polynesiensis* [40] although they can accumulate enough toxin to cause human poisoning [41].

### 2.3. Trophic Level 3 (Blotched Javelin Fish)

To simplify our calculations, we based the food chain conversion of P-CTX-4B to P-CTX-1 (see Lewis and Holmes [9]) on a 1:1 stoichiometry which maximizes the modelled transfer of toxin, although losses are likely in the biotransformation process. For this scenario, we treated the conversion of P-CTX-4B to P-CTX-1 as occurring in blotched javelin fish, because P-CTX-1 was the major toxin found in these fish from Platypus Bay [11], and Ikehara et al. [21] have shown that this biotransformation occurs in fish. However, we do not know the extent to which biotransformations occur in the invertebrates they feed upon.

Based upon an estimated average toxin burden for shrimps of 2.1 × 10^−10^ g/shrimp of P-CTX-4B, it would take the consumption of between 40,000–160,000 shrimps weighing a total of 6–24 kg (average weight 0.15 g [10]) by blotched javelin fish to transfer the required toxin to produce a lowly toxic 10 kg Spanish mackerel (Table 3). These are unrealistically high numbers of shrimps for a single blotched javelin to eat, but it is likely the Spanish mackerel could acquire the toxin from preying upon a toxic school of these small fish. Blotched javelin are the only fish species so far known to occur in toxic schools in Australia [5,10]. Blotched javelin can exceed 25 cm in length, but they commonly occur in Queensland waters to ~12 cm length at which size they start to become sexually mature [42]. If a 10 kg Spanish mackerel acquired the toxin from feeding on three blotched javelins, the average number of toxic shrimps eaten by each blotched javelin fish would reduce to 13,000–53,000 shrimps (Figure 1). However, this still appears an unrealistically high range for a single fish to eat over a short period of time. Lewis and Sellin [11] extracted 920 MU of P-CTX-1 eq. from the pooled flesh of a school of 54 blotched javelins, suggesting an average minimum toxin burden of 17 MU/fish (since toxins from the viscera were not included in the analysis). This would equate to ~5.1 µg P-CTX-1 eq./blotched javelin and suggests that only one to three fish with this level of toxin burden may have to be preyed upon to produce a mildly toxic 10 kg Spanish mackerel (Table 2). This supports the conclusion that shrimps may sometime accumulate much greater levels of toxin than estimated in Table 3, or that additional pathways for transfer of toxins across trophic levels exist in Platypus Bay, or that unknown low toxicity analogs that can biotransform to P-CTX-1 exist in the Platypus Bay food chain. If shrimps can accumulate 10–100-fold higher CTX concentrations than found by Lewis et al. [10], then each blotched javelin would only need to prey upon 130–5300 shrimps to accumulate sufficient toxin for a school of three fish to produce a lowly toxic 10 kg Spanish mackerel. A range of invertebrates other than shrimps also occur within the macroalgal substrate which could facilitate the transfer of CTX, including polychaetae and nematode worms, crabs, and small gastropod snails, with the latter often being the most visually obvious; however, in the limited sampling to date, no CTX could be detected in these alternative prey species [10].

### 2.4. The Substrate Supporting Populations of Gambierdiscus in Platypus Bay

The highest known population density of *Gambierdiscus* from Platypus Bay (556 cells/g [8]) is consistent with the most common range (100–1000 cells/g) for cell densities reported from the Pacific and Atlantic Oceans [13]. Much higher densities are yet to be discovered from Platypus Bay, although the largest population densities known from Australia are 8,255 cells/g from New South Wales [43] and 2000 cells/g from Queensland [44]. Only a very small proportion of the known global distribution of *Gambierdiscus* densities exceed 10,000 cells/g [13]. While a hypothetical cell density an order of magnitude greater than modelled (~5000–6000 cells/g) may increase the potential for more *Gambierdiscus* to be consumed by invertebrates within the three-dimensional space of the macroalgal layer, and therefore the probability for more toxin to be transferred to higher trophic levels, it does not change the number of *Gambierdiscus* required to produce a toxic fish unless cell toxicity also changes.

We can estimate the amount of macroalgal substrate (*Cladophora*) supporting the *Gambierdiscus* populations that would have to be processed through the second trophic level of the marine food chain in Platypus Bay to produce a lowly toxic 10 kg Spanish mackerel (Table 3). This estimate is based upon the maximum population density of *Gambierdiscus* (556 cells/g) currently known for Platypus Bay [8]. The amounts of macroalgal substrate needed to support the ciguateric food chain in Platypus Bay to produce a single lowly toxic fish (22–88 kg, Table 3) appears to be considerable; although, this would reduce if higher population densities or more toxic cells of *Gambierdiscus* occur. We are not aware of any similar data in the literature for us to compare the amount of substrate needed to produce toxic fish. However, knowing the amount of substrate that produces a toxic fish could help design risk mitigation strategies. For example, if remote sensing techniques could be developed to monitor the biomass of *Cladophora* in Platypus Bay, which we know changes significantly through time [5], then it could be used in conjunction with monitoring of the toxicity and abundance of *Gambierdiscus* to estimate the amount of substrate that would need to be removed from Platypus Bay through trawling or dredging [5,9] to reduce the likelihood of production of toxic Spanish mackerel.

## 3. Implications and Conclusions

Our model is a first attempt to conceptualize the scale of *Gambierdiscus* populations needed to be assimilated and transferred through marine food chains to produce a ciguateric fish. Our model suggests that large cell populations are required to produce a single toxic fish, which along with the capacity for fish to lose toxicity over time through depuration [5], explains the patchiness and relative rarity of ciguatera in Australia, both in space and time. The precision of this modelling can be improved by research to better define rates for toxin production, biotransformation, and food chain assimilation. Research is also required to identify and quantify the CTX produced by Australian *Gambierdiscus*. We do know that toxicity of the *Gambierdiscus* growing in Platypus Bay is often much lower than we have used in our modelling [8], suggesting that at times it takes even larger cell populations to produce a toxic Spanish mackerel, and/or that populations likely cycle through periods of high and low toxicity, as shown recently for *Gambierdiscus* in the Caribbean [45]. Indeed, such periods of low CTX production likely correlate with periods of low ciguatera risk from blotched javelin fish and Spanish mackerel. We have known for decades that low population densities of *Gambierdiscus* are a common component of benthic macroalgae, turf algae, and biodetritus along the east coast of Australia [43,44,46,47,48,49], indicating that the presence of such populations on their own are not a good predictor for ciguatera risk.

We have modelled the trophic transfer of CTX to produce a lowly toxic 10 kg Spanish mackerel with a flesh concentration of 0.1 µg/kg of P-CTX-1. The P-CTX-1 concentrations so far measured from Spanish mackerel that have caused poisonings range between 0.1–1.0 µg/kg P-CTX-1 [4], although this data is from a small number of samples. We suspect that Spanish mackerel may sometimes accumulate higher concentrations, as suggested by the death of an otherwise healthy female patient attributed to eating toxic Spanish mackerel fillets [50]. Modelling ciguatera food chains for coral reef fishes from the Great Barrier Reef will likely be more challenging as these fishes appear to have a lower risk of causing ciguatera than Spanish mackerel on the east coast of Australia [5].

In Platypus Bay, the transfer of CTX between trophic levels 3 (blotched javelin) and 4 (Spanish mackerel) results in a reduction in CTX concentration [5], unless large numbers of contaminated fishes are consumed by apex predators. A similar reduction in toxin concentration appears to also occur for some food chains in French Polynesia [51,52,53,54]. This dilution of CTX concentrations appears contrary to bioconcentration models for the bioaccumulation of CTX in fishes [55]. However, it is consistent with our models’ reduction in total toxin burden for each successive trophic level. Indeed, depuration of CTX as it passes through the food chain [5,20,23,25,56] necessitates additional CTX be produced to enable a high trophic level fish to become poisonous, placing further constraints on the model. In contrast to planktonic food chains, where toxin accumulation in filter-feeders appears to be explained by a combination of known cell toxicities and toxic dinoflagellate concentrations, the scale of toxin burden transferred across each trophic level of our model suggests that simply hypothesizing larger populations of more toxic cells quickly exceeds the known limits of CTX-production by *Gambierdiscus* or requires unrealistically high numbers of toxic prey organisms to occur within a trophic level. These considerations highlight likely major gaps in our knowledge of the production and food chain transfer of CTX, either of significant concentration steps in lower trophic levels, and/or a major underestimation of production of precursors that can biotransform to P-CTX-1. Based upon current knowledge, and the levels of toxin suggested by our model to produce a mildly toxic Spanish mackerel, it seems surprising that any fish accumulates sufficient toxin to become poisonous, let alone accumulate levels of ciguatoxin lethal to humans [31,50].

While modelling of *Gambierdiscus* populations is not new [57,58], this is a first attempt to quantitatively model the production and flux of CTX through a specific food chain to produce a toxic fish. Because of the limited available data, our estimates for the number of Platypus Bay *Gambierdiscus* to produce a toxic Spanish mackerel are quite broad, ranging from 12–49 million cells for a 10 kg fish. In addition, our estimates for the number of toxic shrimps that need to be consumed to transfer CTX to the third trophic level may be unrealistically high, suggesting that the parameters controlling the production and flow of CTX to Spanish mackerel in Platypus Bay need further research. It is also possible that far greater population densities of *Gambierdiscus* sometimes occur on the *Cladophora* than sampled to-date, and/or more toxic *Gambierdiscus* species/strains sometimes occur in Platypus Bay, and/or, the assimilation rates for toxin transfer across trophic levels are more efficient than assumed by our model. It is likely that many reef communities have shorter food chains for the transfer of CTX than those in Platypus Bay, which may increase the efficiency of toxin transfer in these ecosystems but would reduce the number of potential concentrating steps. Research is needed to better define the kinetics of toxin transfer which could ultimately lead to the development of quantitative ecological models that could be used to reduce the risk of ciguatera.

## 4. Materials and Methods

Our modelling uses previously published data to quantify the flow of ciguatoxins (CTX) across four trophic levels of a marine food chain in Platypus Bay, K’gari (Fraser Island), Queensland, Australia, an ecosystem previously assessed for the accumulation and depuration of CTX [5]. The apex predator of this ciguateric food chain is the Spanish mackerel (*Scomberomorus commerson*), one of the highest risk fish species for ciguatera on the east coast of Australia [2,3,4,5]. The major CTX that accumulates in Spanish mackerel is the most toxic of the known analogs, Pacific-ciguatoxin-1 (P-CTX-1, also known as CTX-1B) [11,12]. We developed our model by choosing a target concentration of 0.1 µg/kg of P-CTX-1 (0.1 ppb) in the flesh of a 10 kg Spanish mackerel. The quantity of toxin required to be transferred through each trophic level to cause this level of contamination was then back calculated using three different rates of toxin assimilation. This model incorporates the production of the less toxic P-CTX-1 precursors P-CTX-4A (CTX4A) and -4B (CTX4B) by *Gambierdiscus* sp., and then the transfer and biotransformation of these in the Platypus Bay food chain to contaminate Spanish mackerel.

## Figures and Tables

**Figure 1 toxins-14-00534-f001:**
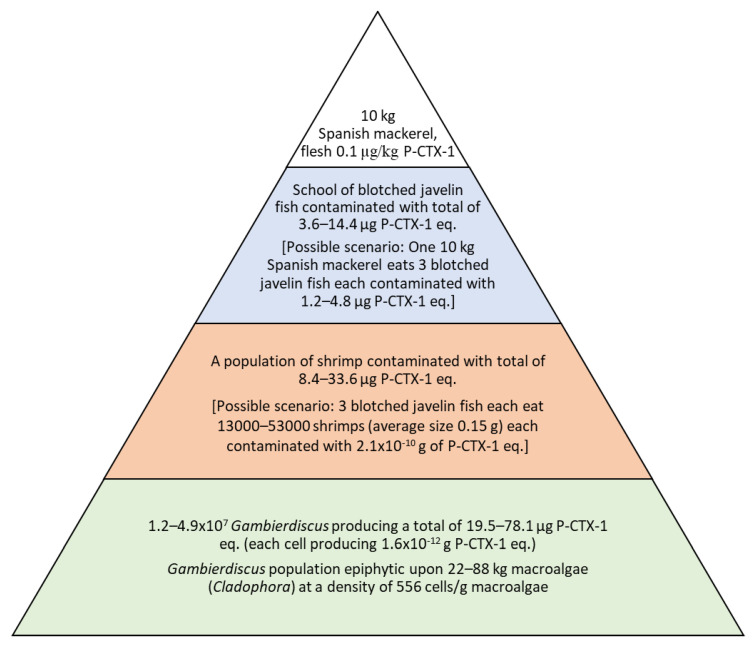
Modelled trophic pyramid producing a mildly ciguateric 10 kg Spanish mackerel (*Scomberomorus commerson*) in Platypus Bay, Australia.

**Table 1 toxins-14-00534-t001:** Total P-CTX burden in Spanish mackerel with a muscle (flesh) concentration of 0.1 µg/kg of P-CTX-1 ^1^.

Spanish Mackerel (kg)	Flesh (kg) Estimated from Equation (1)	Flesh P-CTX-1 (µg) Burden at 0.1 µg/kg	Total P-CTX-1 (µg) Burden in Fish (40% of Toxin Burden)	Total P-CTX-1 (µg) Burden in Fish (10% of Toxin Burden)
10	6.2	0.62	1.6	6.2

^1^ assuming flesh contributes 10–40% of total P-CTX burden by weight [25].

**Table 2 toxins-14-00534-t002:** Estimated P-CTX burden in Platypus Bay trophic levels required to contaminate a 10 kg Spanish mackerel muscle with 0.1 µg/kg P-CTX-1.

Modelled Assimilation Efficiency ^1^	Trophic level 4: Target P-CTX (µg) Burden in Spanish Mackerel ^2^	Trophic Level 3: Required P-CTX (µg) Burden in Blotched Javelin Fish	Trophic Level 2: Required P-CTX (µg) Burden in Shrimps	Trophic Level 1: Required P-CTX (µg) Burden in *Gambierdiscus*
6%	1.6–6.2	25.9–104	431–1,720	7,190–28,750
43%	1.6–6.2	3.6–14.4	8.4–33.6	19.5–78.1
100%	1.6–6.2	1.6–6.2	1.6–6.2	1.6–6.2

^1^ assimilation efficiency of 6.1% reported for P-CTX-1 in juvenile grouper [23], 43% for C-CTX in pinfish [25], or 100% (no loss of toxin). ^2^ assuming flesh contributes 10–40% of total P-CTX burden by weight (data from Table 1).

**Table 3 toxins-14-00534-t003:** Estimated shrimp (trophic level 2), *Gambierdiscus* (trophic level 1), and supporting macroalgal substrate (*Cladophora*) required to contaminate Spanish mackerel muscle at 0.1 µg/kg P-CTX-1 ^1^.

Spanish Mackerel (kg)	Trophic Level 2: P-CTX (µg) Burden in Shrimps to Contaminate Trophic Level 3	Trophic Level 2: Number of Shrimps ^2^ Required to Contaminate Trophic Level 3	Trophic Level 1: *Gambierdiscus* Cells ^3^ Required to Contaminate Trophic Level 2	*Cladophora* Substrate (kg, Wet Weight) to Support 1.2–4.9 × 10^7^ *Gambierdiscus* ^4^
10	8.4–33.6	40,000–160,000	1.2–4.9 × 10^7^	22–88

^1^ data for total toxin burden at each trophic level from Table 2, assuming 43% assimilation efficiency. ^2^ assuming 2.1 × 10^–10^ g of P-CTX/shrimp. ^3^ assuming 1.6 × 10^–12^ g P-CTX/cell. ^4^ estimated from Holmes et al. [8].

## Data Availability

Data sources are attributed in the article.

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
