# Peer review of "Origin of Ciguateric Fish: Quantitative Modelling of the Flow of Ciguatoxin through a Marine Food Chain"

_toxins, 2022, doi:10.3390/toxins14080534_

Round 1

Reviewer 1 Report

The manuscript “Modelling bioaccumulation of ciguatoxin through a marine food chain” addresses a topic of interest concerning the impact of food chain dynamics on ciguatera risk. Authors use a model which is a first attempt to conceptualize the scale of Gambierdiscus populations needed to be assimilated and transferred through marine food chains to produce a ciguateric fish.

This work can help future researchers identify key knowledge gaps to improve understanding of how CTX flows through marine food chains to cause human poisoning. This model suggests that large cell populations of CTX-producing Gambierdiscus are required to produce a single toxic fish, which explains the relative rarity of ciguatera in Australia.

While modelling of Gambierdiscus populations is not new, this work is a first attempt to quantitatively model the production and flux of CTX through a specific food chain to produce a toxic fish.  The parameters controlling the production and flow of CTX to Spanish mackerel in Platypus Bay need further research but the authors clearly indicate this limitation, which is why I consider the proposed work to be of great scientific honesty.

Author Response

We thank Reviewer 1 for his/her comments, especially for the comment on the manuscript’s scientific honesty. Reviewer 1 has no criticism and suggested no changes to the manuscript. Therefore, we have made no changes to the manuscript.

Reviewer 2 Report

The published data about ciguatoxins must be actualized. Initial reports described an LD50 in mice of 50 µg/kg while other reports have described LD50 up to 1000 µg/kg.

Other major concerns:

There is not model to explain the titles.

As the authors explain in page 2, lines 55-57, the mild symptomos needed for ciguatoxins to produce effects in 2 out of 10 people is ten-fold higher than the recomended limits.... Therefore the manuscript must be actualized.

From table 2, it is not clear that the assimilation eficiency from 6 to a 100% will be between 1.6 and 6.2 for trophic level 4, and the same for trophic leves 1 to 3 and 100% of assimilation efficiency.

The matherial and methods section is not present.

Author Response

Reviewer comment: The published data about ciguatoxins must be actualized:

Authors response: It is not clear what the Reviewer means by “actualized”? We have assumed he/she means verified experimentally. However, this is a theoretical paper modelling the transfer of ciguatoxins using the best available published data. It provides context to help future researchers design experiments which will experimentally prove or disprove the conclusions from our modelling. So, we agree that the model outputs and our conclusions should be tested experimentally in the future, but as acknowledged by the other Reviewers, this manuscript is the first to quantitatively model transfer of ciguatoxins through a marine food chain and provides scope and context for this process. We have outlined the scope of this theoretical paper in lines 63-72 as well as in the newly incorporated Materials and methods section (lines 363-377). Therefore, we have made no change to manuscript (other than inclusion of a Materials and methods section).

Reviewer comment: Initial reports described an LD50 in mice of 50 µg/kg while other reports have described LD50 up to 1000 µg/kg.

Authors response: The intraperitoneal (i.p.) toxicity in mice to different ciguatoxin analogs varies as suggested by the reviewer; however, Pacific ciguatoxin analogs have only been shown to range between 0.32-5.9 µg/kg (FAO 2020), with toxic doses up to 1000 µg/kg only being shown for oral toxicity of some analogs (such as PCTX3C). However, this manuscript focuses on P-CTX-1, which has been shown to be the major ciguatoxin in most fish that cause ciguatera in Australia, and in the Platypus Bay marine food chain that we model in the manuscript. P-CTX-1 has an i.p. LD50 in mice of ~0.3 µg/kg (range 0.25-0.35 µg/kg, n=4, FAO 2020) and has never been shown to have an i.p. LD50 up to 1000 µg/kg. We only model P-CTX-1 and its precursors because this is the major ciguatoxin in Spanish mackerel from the east coast of Australia which is the focus for this study (references throughout manuscript). Therefore, we have made no change to the manuscript.

Reviewer comment: There is not model to explain the titles.

Authors response: It is not clear what the Reviewer means by this comment, but we assume he/she means that the title does not adequately describe the model outcome (because toxin does not necessarily bioaccumulate through trophic levels). On reflection, we agree with the Reviewer and have changed the title accordingly. We have also deleted the word bioaccumulation on line 50 to reflect this change.

Reviewers comment: As the authors explain in page 2, lines 55-57, the mild symptomos needed for ciguatoxins to produce effects in 2 out of 10 people is ten-fold higher than the recomended limits.... Therefore the manuscript must be actualized.

Authors response: We do not see the logical connection between these two sentences. As we have pointed out above, this is a theoretical paper utilizing published data to predict outcomes which need to be experimentally verified, modified, or disproved. Therefore, we have made no change to the manuscript.

Reviewers comment: From table 2, it is not clear that the assimilation efficiency from 6 to a 100% will be between 1.6 and 6.2 for trophic level 4, and the same for trophic levels 1 to 3 and 100% of assimilation efficiency.

Authors response: The Reviewer’s concern is not clear to us, but we assume he/she is asking for clarification about how the data in Table 2 is calculated. Footnote 2 in Table 2 explains that the data for Trophic level 4 is taken from Table 1 and was calculated based upon the published data of Bennett and Robertson (2021) who found that the flesh of pinfish contributed between 10%-40% of the total ciguatoxin load of the fish. To improve clarity, we have added the word “data” to footnote 2 (line 121).

Alternatively, the Reviewer could be concerned about interpreting the legend of the first column in Table 2 which lists the assimilation efficiencies we have used in our modelling. To improve clarity, we have changed the title of the column showing the 3 assimilation efficiencies from “Assimilation efficiency1” to “Modelled assimilation efficiences1”.

Reviewers comment: The matherial and methods section is not present.

Authors response: Our model is based on back calculating a total toxin load from a hypothetical flesh toxin concentration for a 10 kg Spanish mackerel and then using published assimilation rates to calculate the transfer of toxin through each trophic level. The calculations are outlined and discussed in a step-by-step manner through the manuscript, as the calculations for each subsequent table are derived from the preceding tables. However, in response to the Reviewers’ concern, we have included a Materials and methods section that describes the process for the modelling (lines 363-376.

Reviewer 3 Report

This paper describes quantitative modeling of the transfer of CTX between four trophic levels of Platypus Bay.   The authors developed their model by choosing a target CTX concentration for the flesh of Spanish mackerel and then back calculating the quantity of toxin required to be transferred through each trophic level to cause this level of contamination. This approach can help future researchers identify key knowledge gaps to improve understanding of how CTX flows through marine food chains to cause human poisoning. This model suggests that large cell populations of CTX-producing Gambierdiscus are required to produce a single toxic fish, which along with the capacity for fish to lose toxicity over time through depuration, explains the relative rarity of ciguatera in Australia.  The results shown in this paper are interesting to researchers in the field of toxins as well as food chemistry.  Thus, I recommend the publication in Toxins.   Aside from the recommendation, the following should be revised before publication.

1)    Although the authors use the nomenclature for CTX, like P-CTX-1 and P-CTX-4B, I strongly recommend they should use the nomenclature that Yasumoto originally used, like CTX1B and CTX4B.  The nomenclature like CTX1B and CTX4B is also used in the “report of the expert meeting on ciguatera poisoning” published by FAO and WHO.

Author Response

Reviewers commentAlthough the authors use the nomenclature for CTX, like P-CTX-1 and P-CTX-4B, I strongly recommend they should use the nomenclature that Yasumoto originally used, like CTX1B and CTX4B.  The nomenclature like CTX1B and CTX4B is also used in the “report of the expert meeting on ciguatera poisoning” published by FAO and WHO. 

Authors response: The nomenclature proposed in the report of the expert meeting on ciguatera poisoning (FAO 2020), has not been universally accepted or adopted, e.g., these recently published papers have not followed the FAO 2020 nomenclature: Holmes et al. 2021 Toxins 13, 515. https://doi.org/10.3390/toxins13080515, Habibi et al. 2021 Toxins13, 525. https://doi.org/10.3390/toxins13080525, Leist 2021 ALTEX 38(1), 177-182. doi:10.14573/altex.2012311, Pierre et al. 2021 Mar. Drugs 19, 387. https://doi.org/10.3390/md19070387, MacDonald et al. 2021 BRAIN 144; 1711–1726 doi:10.1093/brain/awab086, Castro et al. 2022 J. Mar. Sci. Eng. 10, 835. https://doi.org/10.3390/jmse10060835, Kryuchkov et al. 2022 Toxins 14, 399. https://doi.org/10.3390/toxins14060399.

While Professor Yasumoto’s contribution to the field is immense, the suggestion that his nomenclature should be used based upon first use, does not conform with precedence as his group where not the first to publish a number or letter attachment to the abbreviation for CTX to indicate an analog. The first such case (CTX-1) was published by Lewis et al. 1991 Toxicon 29 (9) 1115-1127 as acknowledged in the Table of synonyms in FAO 2020 (Chapter 3, Table 5).

However, to avoid any confusion of toxin identity, we have included the CTX nomenclature from FAO (2020) in the first use throughout the manuscript, and in the newly included Materials and methods section (i.e., lines 12, 50, 130, 160, 161, 179, 370, 375).

Reviewer 4 Report

Authors reported the modeling of ciguatoxin accumulation through a food chain.

Although there are a lot of  ciguatoxin analogs, authors focused on the accumulation of p-CTX-1 in fish to simplify the model. 

p-CTX-1 is the most potent toxin among ciguatoxins.

Therefore, toxicity of fish may be estimated.

I think this manuscript is worthy being published in toxins.

Author Response

We thank the Reviewer for his/her comments. Reviewer 4 has no criticism or suggested changes to the manuscript. Therefore, we have made no changes to the manuscript in response.

Round 2

Reviewer 2 Report

the authors have addressed all the comments